# Fluorescence-Guided Surgery for Gliomas: Past, Present, and Future

**DOI:** 10.3390/cancers17111837

**Published:** 2025-05-30

**Authors:** Benjamin Rodriguez, Cole S. Brown, Jhair Alejandro Colan, Jack Yin Zhang, Sakibul Huq, Daniel Rivera, Tirone Young, Tyree Williams, Varun Subramaniam, Constantinos Hadjipanayis

**Affiliations:** 1Department of Neurosurgery, Icahn School of Medicine at Mount Sinai, New York, NY 10029, USA; cole.brown@icahn.mssm.edu (C.S.B.); jack.zhang@icahn.mssm.edu (J.Y.Z.); daniel.rivera@icahn.mssm.edu (D.R.); tirone.young@icahn.mssm.edu (T.Y.); varun.subramaniam@icahn.mssm.edu (V.S.); 2Sinai BioDesign, Department of Neurosurgery, Mount Sinai, New York, NY 10029, USA; williams.d.tyree@gmail.com; 3Department of Neurological Surgery, University of Pittsburgh Medical Center, Pittsburgh, PA 15213, USA; colanja2@upmc.edu (J.A.C.); huqs5@upmc.edu (S.H.); 4Department of Biomedical Engineering, Rensselaer Polytechnic Institute, Troy, NY 12180, USA; 5Brain Tumor Nanotechnology Laboratory, UPMC Hillman Cancer Center, Pittsburgh, PA 15232, USA; 6Center for Image-Guided Neurosurgery, Department of Neurological Surgery, University of Pittsburgh School of Medicine, Pittsburgh, PA 15213, USA

**Keywords:** fluorescence-guided surgery, glioblastoma, 5-ALA, intraoperative imaging, high-grade glioma

## Abstract

Glioblastoma is an aggressive brain tumor with a poor prognosis, partly due to the challenge of completely removing all tumor cells during surgery. Fluorescence-guided surgery (FGS) is a technique that helps surgeons see tumor cells more clearly by using special dyes that light up under certain types of light. This approach has improved how much of the tumor can be removed safely, which can lead to better patient outcomes. In this paper, we review the history, current practices, and future possibilities for using FGS in brain tumor surgery, with a focus on glioblastoma. We discuss the different fluorescent agents being used and the technologies that enhance their effectiveness. Our goal is to provide a comprehensive overview of FGS, highlight recent advancements, and explore how further innovations could expand its use in neurosurgery, potentially improving patient survival and quality of life.

## 1. Introduction

Glioblastoma, the most common primary malignant brain tumor, encompasses 50.9% of all malignant central nervous system (CNS) diagnoses and has a median survival of 15 months with the current standard of care [1,2]. This prognosis has remained dismal over decades due to the high tumor-recurrence rate, which is driven by residual and/or resistant tumor cells in the infiltrative margin that are left behind after resection [3,4]. For this reason, maximizing the extent of resection (EOR) and consequent cytoreduction is an integral component of standard of care, with a demonstrated survival benefit [5,6]. There is therefore great interest in efforts to improve intraoperative visualization of glioblastoma, particularly at the infiltrative edge, in order to maximize EOR [7,8]. One of the most significant innovations in this realm is the advent of fluorescence-guided surgery (FGS), which has revolutionized intraoperative visualization and improved the extent of resection rates in patients with glioblastoma, the most common high-grade glioma (HGG) [9,10,11,12].

FGS uses photoactive agents called fluorophores to illuminate structures of interest during surgery. These fluorophores absorb a range of wavelengths and emit a specific wavelength, with these excitation and emission wavelengths typically in the visible to near-infrared (NIR) spectrum (500–900 nm) [13,14,15]. Fluorescence in this spectrum is utilized to provide contrast between a target of interest and surrounding tissue. Fluorescence guidance is used throughout the medical field, from identifying native hepatobiliary anatomy to photodynamic diagnosis of bladder cancer to assessing coronary artery bypass graft patency and perfusion. In neurosurgery, FGS is predominantly used within vascular neurosurgery (visualizing blood vessels, such as in aneurysm and bypass surgery) and neurosurgical oncology, in which it is now approved and standard of care for glioblastoma surgery.

As FGS continues to advance, distinguishing between clinically established and experimental applications is essential. Experimental applications generally include investigational usage in existing clinical trials, off-label applications, or novel technologies absent of wide-scale validation. Clinically established uses are interpreted as those with regulatory approval and demonstrated safety and efficacy in phase III clinical trials or diffuse clinical application. In this review, we discuss historical perspectives, current practices, and future directions for FGS in neurosurgical oncology, with a focus on glioblastoma.

## 2. Historical Perspective

The use of fluorescence in the medical sciences began in the late 19th century, with the word “fluorescence” first entering the lexicon in 1852 [16,17]. The first fluorescent stain, fluorescein, was invented in 1871 [16,18]. Decades later, in 1947, it was discovered that fluorescein could be used to differentiate neoplastic and normal tissue during brain tumor surgery [19]. Intracranial lesions demonstrated this quality most consistently, with tumor tissue emitting robust yellow fluorescence upon exposure to ultraviolet (UV) light [19]. Unsurprisingly, neurosurgery became the first surgical field to explore the utility of intraoperative fluorescence when, the following year, neurosurgeons at the University of Minnesota Medical School administered intravenous fluorescein to patients undergoing craniotomy for suspected brain tumors [20]. They reported that the fluorescing tissue was indeed found to be neoplastic upon neuropathological evaluation in 44 of 46 patients [21]. In the decades since, various fluorophores have been developed and studied for application in FGS, the most popular today being 5-aminolevulinic acid hydrochloride (5-ALA), indocyanine-green (ICG), and fluorescein. While 5-ALA is the only Food and Drug Administration (FDA)-approved agent specifically for glioma surgery in the United States [12,22], ICG and fluorescein are FDA approved for use in ophthalmologic and vascular procedures but remain off-label for neurosurgical tumor resection. In Europe, fluorescein has received CE marking and is more widely adopted for glioma surgery. The study that galvanized the eventual approval of 5-ALA was a landmark multicenter, phase III clinical trial from Dusseldorf, Germany, using 5-ALA-guided resection of malignant gliomas in 270 adult patients [12,23]. The control group was standard microsurgery using the optical microscope with white light, and the primary endpoint was the number of patients with residual enhancing tumor after resection. The study found that the tumor was resected completely (no contrast-enhancing residual tumor at 72 h post-resection) in 65% of the 5-ALA group compared to 36% of the control group. A timeline of key historical events in the advancement of FGS for neurosurgery is provided in Figure 1.

## 3. Fluorophores

### 3.1. Indocyanine Green (ICG)

ICG is the most commonly used fluorophore in surgery [20,24]. It is a water-soluble compound that is delivered intravenously in a reconstituted aqueous solution that binds serum albumin at the hydrophobic alpha- and beta-lipoprotein molecules [25,26,27,28]. These properties make ICG a great candidate for angiography as it distributes evenly and rapidly in the blood [29,30]. ICG is excited by wavelengths in the range of 750–800 nm, and it emits a light that peaks at 832 nm [29,31].

ICG was originally employed as a quantitative measurement of hepatic and cardiac function [32,33]. The fluorophore was used to quantify the amount of albumin in the serum of patients by measuring the decay of fluorescence. Albumin-bound ICG takes longer to decay; therefore, faster decay of fluorescence represents lower serum albumin levels [32]. ICG was later adopted into ophthalmology for infrared fundus angiograms and also as an intraoperative contrast agent to assist in performing vitrectomies to treat macular holes [34,35,36,37].

In today’s medical practice, ICG is used in angiograms throughout the body. In general surgery, it is used to evaluate the integrity of surgical anastomosis in esophageal, gastric, small bowel, and colorectal surgery [38,39,40,41,42,43]. In neurosurgery, ICG is most commonly used as a contrast agent for ICG angiography (ICGA) during aneurysm surgery and to illuminate blood vessels during tumor surgery [44,45]. An example of cutting-edge ICG technology is the SPY Elite System (Stryker, Kalamazoo, MI, USA), a handheld, laser-assisted ICGA that has been used to monitor reperfusion of skin flaps in plastic surgery and track sentinel lymph nodes for breast cancer [46,47,48,49]. The University of Pennsylvania has also been productive in the ICG space, especially in the development of the second window technique, which will be described in depth below [50].

### 3.2. Fluorescein

Fluorescein sodium dye is an organic compound that is readily soluble in water and most often used in ophthalmology [51,52,53]. The aqueous solution is excited by light in the cobalt visible spectrum (465–490 nm) and fluoresces as bright green in the 520–530 nm range [54]. In ophthalmology, fluorescein can be applied directly to the eye using a stained paper strip; however, it is typically administered orally or intravenously [55].

Fluorescein dye was first used in humans to perform ophthalmological angiograms in 1959 [56]. Since then, the use of fluorescein in ophthalmology has expanded to assist in the “tear breakup time test” to quantify dry eyes, but it is still most applicable for retinal vasculature angiography [52,53]. In other disciplines of medicine, Fluorescein has been adopted to provide an assist in intraoperative imaging of peripheral nerves and colorectal epithelium [57,58]. Diagnostically, it is used to quantify occlusion of myocardial microcirculation for evaluation of CAD [59].

As far as its use in neurosurgery, intrathecal fluorescein is used to identify and localize CSF leaks during skull base surgeries or in the case of traumatic CSF leaks [60]. Additionally, it can be used for the identification of high-grade gliomas, although 5-aminolevulinic acid is more sensitive and specific. Lastly, fluorescein has been used to assess cerebral blood flow during aneurysm clipping, AVM resection, and other neurovascular procedures [61].

### 3.3. 5-Aminolevulinic Acid (5-ALA)

5-ALA is the newest fluorophore in clinical use and became FDA-approved for neurosurgical applications in 2017 [62,63]. 5-ALA is an amino acid that occurs naturally and serves as a precursor to heme and protoporphyrin IX (PpIX) [64]. The accumulation of PpIX in neoplastic tissue is directly responsible for the fluorescent capabilities of 5-ALA. Multiple studies have confirmed the high diagnostic accuracy of 5-ALA-induced fluorescence in HGGs. The accumulation of PpIX in HGGs results in an unprecedented high sensitivity and positive predictive value of fluorescence visualization correlating with tumor tissue [65,66,67]. One multicenter study completed in the US confirmed a positive predictive value of over 96% [65]. Initial use of 5-ALA as a fluorophore in glioma surgery was pioneered by Stummer et al. in 1998, demonstrating intraoperative tumor fluorescence in malignant gliomas [68]. This foundational work led to a pivotal multicenter, randomized Phase III trial published in 2006, in which 5-ALA-guided surgery achieved complete resection of contrast-enhancing tumor in 65% of patients. The 5-ALA group also experienced significantly improved 6-month progression-free survival (41% vs. 21%) [12]. Since then, FGS with 5-ALA has gained considerable popularity among neurosurgeons and is now the standard of care for glioblastoma surgery at many centers around the world [63].

5-ALA is administered orally and absorbed through the gastrointestinal tract into the bloodstream; it is rapidly metabolized in brain tumors, where it is converted to PpIX in the heme biosynthesis pathway [69]. PpIX accumulates within brain tumor cells and is excited by blue light (400 nm range), leading to red-violet fluorescence at 635 nm and 704 nm emission peaks [69,70]. Intraoperatively, normal brain parenchyma appears blue, while the PpIX accumulated in the solid tumor bulk tissue appears red and the tumor margin pink [71]. In addition to the original Stummer study, multiple other studies have confirmed the greater ability to more completely resect tumors. In a meta-analysis performed by Ejamel et al., a gross total resection rate (removal of at least 98% of contrast-enhancing tumor) of 75.4% (418/565 patients) was achieved with 5-ALA FGS [72]. More recently, a randomized multicenter trial in France confirmed in a comparison between 5-ALA FGS and conventional microsurgery that greater overall tumor resection could be performed with 5-ALA FGS in GBM patients [73]. These data highlight the utility of 5-ALA FGS in the resection of HGG and overall better experiences in the OR for neurosurgeons.

5-ALA is also being implemented in fields other than neurosurgery, including in other oncologic surgeries for dermatology, head and neck, gynecology oncology, and orthopedics. Filip et al. described the use of 5-ALA for resection of head and neck squamous cell carcinoma [74]. Bickels et al. demonstrated in a study of 24 patients that 5-ALA can help decrease the likelihood of tumor recurrence in the surgical removal of desmoid tumors, solitary fibrous tumors, and dermatofibrosarcoma protuberans [75]. There are also promising preclinical in vivo models studying the effectiveness of 5-ALA in bone cancers such as chondroblastoma [76]. An overview of all described fluorophores can be found within Table 1.

## 4. Practical Clinical Application

Since its FDA approval in 2017, 5-ALA has been regularly implemented in the resection of HGG in the US. 5-ALA is administered orally anywhere from 2 to 4 h prior to surgery at a dose of 20 mg/kg body weight [12]. This provides ample time for the preoperative setup, anesthesia, and craniotomy so that peak intraoperative fluorescence will occur during tumor resection. The dosing is based upon experiments in rodents in which the peak fluorescence was observed 6 h after administration [77]. While Stummer et al. demonstrated effective administration and strong fluorescence with their protocol, Kaneko et al. found in a prospective study of 68 patients that maximal fluorescence was observed 7 to 8 h after administration of 5-ALA, with weaker fluorescence lasting as long as 8 to 9 h [12,78]. Others have also confirmed that 5-ALA tumor fluorescence may be better more, than 4 h after administration, with tumor fluorescence reported up to 24 h after administration [79]. The 2 to 4 h time window is the FDA-approved administration criterion, but it is important to further investigate if peak fluorescence extends further so that intraoperative fluorescence can be utilized despite inevitable logistical issues and delays in surgery [80]. Regardless, 5-ALA provides a suitable window for surgeons to clearly visualize tumor tissue in patients with HGG.

The intensity of red fluorescence correlates with the density of tumor cells [71]. The neurosurgeon then decides how much tissue can be safely resected. 5-ALA has excellent sensitivity for detecting tumors, but it has slightly lower specificity; in some cases, fluorescent tissue may therefore contain both tumor tissue and functional brain tissue. Moreover, fluorescence penetration can be limited to approximately 1–2 mm, and signal intensity can vary due to tumor heterogeneity. False positives may occur, particularly in areas of necrosis or inflammation. As a result, surgeons must therefore use clinical judgment in using FGS to guide resections, along with other intraoperative adjuncts such as neurophysiological monitoring [63].

While fluorescence-guided surgery with 5-ALA and other agents is generally well tolerated, it is important to be aware of associated adverse effects. 5-ALA can cause photosensitivity reactions, requiring patients to avoid direct sunlight or strong indoor lighting for 24 h postoperatively. Fluorescein has been linked to mild allergic responses and can cause temporary skin or urine discoloration [21]. Indocyanine green, although rare, has been associated with anaphylactic reactions, particularly in iodine-sensitive individuals [30]. While the incidence of serious adverse events is low, preoperative screening and patient counseling remain essential components of safe clinical practice.

### 4.1. Complementary Technologies

Since the FDA approval of 5-ALA for brain tumor resection, several technologies have been developed to complement and improve the capabilities of FGS. Fluorophore visualization utilizes a short-pass filter (~400 nm) to generate light to excite the fluorophore and a long-pass filter (~650 nm) to detect emitted light from the fluorophore [73,81]. The technologies that assist in the visualization of fluorescence can be widely separated into two field of view (FOV) categories: wide field of view (WFOV) and narrow field of view (NFOV) [82]. WFOV refers to technologies that illuminate the entire surgical cavity and broadly detect the emission of the fluorophore. This category includes the widely used visualization technologies like optical microscopes, loupes, and exoscopes, but with added filters. NFOV illumination focuses on a smaller field, which enhances resolution and allows for quantitative measurement of fluorescence at a cellular scale [83,84,85,86,87,88].

Optical microscopes were the first technology outfitted for compatibility with fluorescence, as described by Stummer and colleagues when they used an optical microscope with an excitation filter at 375–440 nm and an emission filter at >455 nm to visualize malignant gliomas with 5-ALA.

More recently, the three-dimensional (3D) exoscope has gained popularity in the neurosurgical operating room (OR). Exoscopes are 3D high-definition camera systems that project onto a monitor in a neurosurgical OR to provide the entire staff with a visual of the surgical field [89,90,91,92,93,94]. The exoscope provides superior magnification and tumor tissue clarity with the use of LED lighting and a longer working distance and depth of field in comparison to the conventional operating microscope. The use of the exoscope for resection of GBM tumors has been reported with a high extent of tumor resections achieved [95]. These systems also have FGS compatibility to excite 5-ALA, fluorescein, and ICG [96,97]. The exoscope may have both visualization and ergonomic benefits; Dell Pepa et al. recorded ten procedures, five with an exoscope and five with an optical microscope [98]. They found that exoscopes provided superior visualization of vessels, parenchyma, surgical instruments, and fluorescence under the 5-ALA blue filters compared to the optical microscope. The exoscope was also better integrated into surgeon workflows, with less frequent blue-to-white light switching as compared to the optical microscope.

The last of the WFOV devices are FGS-compatible loupes. Multiple commercial systems exist. The REVEAL (Designs for Vision) loupes are equipped with a TriBeam light source that combines a light source with 420 nm and 475 nm filters for excitation and emission of 5-ALA [99,100]. Giatini-Larsen et al. presented a comparison of visualizing fluorescence in three patients using three different modalities: a blue flashlight with an optical microscope, a low-cost headlamp with an optical microscope, and the FGS-compatible loupes alone. The team found fluorescence with all three modalities, but the loupes outperformed the microscope in terms of visualization capabilities [100]. Zhang et al. recorded their experience using the FGS-compatible loops on a cohort of 11 consecutive patients and rated their experience with the loupes as excellent, without any difficulties incorporating the loupes into their workflow [99].

In contrast to WFOV visualization, LFOV optics can detect light emitted in a much smaller pixel, even down to a cellular level [82,101]. Intuitively, this higher resolution protects against the distorting effects of photobleaching from surrounding light and scattering effects off of tissue [86]. More importantly, it allows for a quantitative measurement of fluorescence as opposed to the subjective visual interpretation of the surgeon [66,84,85,86,87,88,102]. The use of fluorescence spectrometry to quantify the emission of fluorescence was pioneered by Diamond et al., who, building off of previous work defining a photon migration model to quantify fluorescence, developed a single-fiber optic probe for both light source and collection [103,104,105]. Similarly to WFOV, NFOV visualization relies on a blue-light excitation wavelength that is coupled with a broadband light sensor to detect fluorescent emission. However, with NFOV, both of these are localized on a much smaller probe [84].

Since their invention, these probes have been adopted by neurosurgeons for glioma surgery. Stummer and team used the probe to perform “fluorescence biopsies” in human subjects with HGG, quantifying the fluorescence of tissue as either “strong” or “weak” fluorescence prior to resection; the specimens were then histo-pathologically evaluated [66]. The group found that “strong” fluorescence had a 100% positive predictive value for tumor and a strong correlation to higher tumor cell density. “Weak” fluorescence had a 95% positive predictive value for tumor and a strong correlation to medium-to-low tumor cell density. Haj-Hosseini et al. developed a spectroscopy system with pulsed modulation to quantify fluorescence in the resection cavity and at the tumor margin before finishing HGG resection procedures [85]. During tumor resections, they used the spectroscopy system to survey residual malignant cells and were able to quantitatively detect 5-ALA fluorescence in the surgical cavity.

The recent advancements in complementary visualization technologies, whether it is WFOV or NFOV, have served to increase surgeon adoption of FGS and improve patient outcomes. An overview of key intraoperative visualization technology is provided in Figure 2.

### 4.2. Therapeutic Applications

In addition to visualization aiding technologies, fluorophores have been incorporated into intraoperative therapies, such as photodynamic therapy (PDT) and sonodynamic therapy (SDT). These interventions capitalize on the reactive nature of PpIX to generate reactive oxygen species and kill cancer cells in the brain [106,107]. Therapeutic-oriented trials are examining possible targeted therapeutic effects of fluorophores when combined with light or low-intensity-focused ultrasound therapy (in PDT and SDT). In the case of intraoperative PDT, a different wavelength of light (635 nm) is administered to activate the 5-ALA metabolite, PpIX, to create reactive oxygen species (ROS), which are toxic to the tumor cells [108,109]. A balloon is placed in the resection cavity at the time of surgery with a laser fiber inserted to illuminate the surgical site and target residual tumor cells. A recent study was completed in France utilizing 5-ALA for intraoperative PDT during glioblastoma surgery. The authors reported the feasibility, efficacy, and safety of this approach in combination with standard of care fractionated external beam radiotherapy with concomitant and adjuvant chemotherapy [110]. In their initial 10-patient study, they found no serious adverse events related to the PDT treatment and, more importantly, 40% of patients ended up surviving over 5 years [111].

With sonodynamic therapy (SDT), focused ultrasound is used to target the intracellular PpIX in tumor cells to generate ROS [112]. SDT does not require any open surgery but uses low-intensity-focused ultrasound waves that are applied through the skull. Current clinical studies are underway in the US evaluating the efficacy of 5-ALA SDT in the treatment of recurrent HGG [113,114,115]. SDT is also being investigated for diffuse intrinsic pontine glioma (DIPG) in young adults and children [116]. Two clinical trials exploring PDT for glioblastoma are available in the US [117,118]. One intraoperative PDT trial is open for enrollment at the University of Pittsburgh Medical Center for newly diagnosed GBM patients utilizing the new PDT agent known as Pentalafen. Taken together, there are multiple active and recruiting clinical trials examining the use of fluorophores for the actual therapy of brain tumors underway in the US.

### 4.3. Adoption and Outcomes

Following the European, Asian, Canadian, and US FDA approval in 2017, 5-ALA FGS has become a standard of care during the resection of HGG worldwide. Multiple studies have confirmed 5-ALA FGS to be safe and effective, with minimal side effects. While there is a noted learning curve in the use of 5-ALA FGS for neurosurgeons, there is no question that more complete resections can be performed during resection of HGG tumors [70].

A comprehensive review by Mansouri et al. highlighted the growing body of evidence supporting the use of 5-ALA in glioma surgery, particularly emphasizing the increasing integration of FGS into neurosurgical training programs, facilitating wider adoption [119]. Ferraro et al. conducted a survey of neurosurgeons across Europe and found that 5-ALA was used by 78% of respondents for HGG surgeries, indicating its widespread acceptance in clinical practice [120].

The increasing adoption of FGS has also spurred technological innovations to enhance its utility. Kaneko and Kumagai described the development of quantitative fluorescence imaging systems that can provide more objective assessments of tumor fluorescence, potentially improving surgical decision-making [121]. Additionally, Widhalm et al. reported on the successful integration of FGS with other advanced imaging modalities such as PET, further refining the surgical approach to gliomas [122].

This growing body of evidence and widespread clinical adoption highlight the utility of 5-ALA FGS in the resection of high-grade gliomas, leading to improved extent of resection and potentially better outcomes for patients. As research continues to expand and technology evolves, FGS is poised to remain a cornerstone in the neurosurgical management of gliomas.

## 5. Future Directions

### 5.1. Improved Performance of 5-ALA

As 5-ALA has gained popularity among neurosurgeons in the resection of HGG, it has more recently been applied to other tumors, including meningiomas and pediatric brain tumors [123]. In 2014, Valdes et al. demonstrated excellent utility of 5-ALA in 12 out of 15 patients with meningiomas, with accuracy as high as 90% in differentiating tumor tissue from normal brain [124]. More recently, Wadiura et al. examined the use of 5-ALA in 191 samples from 85 surgeries and found that 5-ALA had a positive predictive value for tumor of 100% [125]. A multicenter study in the US utilizing 5-ALA for meningioma resection has been completed in the US. In children, Milos et al. demonstrated 5-ALA-induced fluorescence in five patients with low-grade gliomas (LGGs) [126]. LGGs are known not to fluoresce as avidly as HGGs. Another ongoing phase II trial in Germany is investigating the application of 5-ALA in pediatric brain tumors [127]. Given the success of 5-ALA in the resection of glioblastoma in adults, we anticipate seeing more clinical studies on pediatric brain tumor resections using 5-ALA in the coming years.

### 5.2. Exploring Other Fluorophores

Despite its great successes, 5-ALA does have some limitations. Its visible-light emission (635 nm) is significantly absorbed by other endogenous fluorophores, like heme, which can cause interference when identifying tumor margins [22,128]. This flaw is being addressed by incorporating fluorophores with unique emission spectra that do not experience as much endogenous noise. The most promising of these fluorophores is ICG. ICG has an emission peak at 832 nm, which is within the near infrared (NIR) spectrum [29,31,129]. This allows for maximum penetration depth and minimum interference from serosanguinous fluid [129]. Second-Window-ICG (SWIG) is a technique that takes advantage of the permeability of endothelial tissue within peritumoral tissue. A large bolus of ICG is administered, and over time, the ICG leaks through the permeable tumor vasculature and accumulates in the tumor and peritumoral space [22]. The next day, surgery is performed and ICG is visualized under NIR light to identify tumor margins. Lee et al. utilized SWIG in 15 patients by administering 5 mg/kg of ICG 22 h before surgery and found that 12 out of 15 tumors could be visualized using an NIR camera [130]. Using pathology as the gold standard, SWIG ICG had a sensitivity of 98% and a specificity of 45% in identifying tumor tissue. In 2022, Karsalia et al. visualized the glioblastoma resection cavity using an exoscope and SWIG under NIR light to detect residual tumor [131]. The group posited that fluorescence visualized in the cavity would represent residual tumor cells, so all fluorescing tissue was resected. They found that compared to gadolinium-enhanced postoperative MRI scans, SWIG and NIR had a higher sensitivity (96%), a higher negative predictive value (89%), and higher accuracy (91%) in detecting residual tumor.

Fluorescein is another popular fluorophore that has been proposed as an alternative or complement to 5-ALA in glioma resection. Unlike ICG, fluorescein does not address the endogenous fluorophore absorption problem of 5-ALA. Fluorescein fluoresces in the yellow-green spectrum at 520–530 nm, which still faces interference from endogenous fluorophores. However, the benefits of fluorescein lie in its cost-effectiveness and lack of phototoxicity. 5-ALA can be cost-prohibitive and can exhibit phototoxicity and photobleaching in situ [132]. For these reasons, surgeons may choose to use fluorescein off-label in glioma resections [133,134]. This has led to studies into the comparison of effectiveness between 5-ALA and fluorescein. Hansen et al. compared 209 patients who underwent resection for HGGs with 5-ALA (n = 58 patients) or fluorescein (n = 51 patients) and found that there was no statistically significant difference between the two groups in extent of resection, the percent of patients with residual tumor volume less than 0.175 cm^3^, or the median overall survival [132]. The only significant outcome difference was progression-free survival, in which fluorescein outperformed 5-ALA (9.2 months vs. 8.7 months).

Fluorescein has also been tested in conjunction with 5-ALA to improve tumor margin visualization, with the rationale of using 5-ALA to target the tumor and fluorescein to target the peritumoral space [22]. This dual fluorescence has been explored in multiple studies. Molina et al. combined fluorescein and 5-ALA to better visualize gliomas in six patients and concluded that the yellow-green background of fluorescein improved visualization of 5-ALA, which led to improved delineation of the tumor margin [135]. Similarly, Schwake et al. (2015) reported that the dual-labeling approach enhanced the visibility of tumor tissue in areas where 5-ALA fluorescence was weak, particularly at the tumor margins [136]. These studies suggest that combining fluorescein and 5-ALA may provide complementary information, potentially leading to more precise tumor delineation and improved extent of resection.

Emerging molecular strategies are also exploring the use of antibody–fluorophore or antibody–drug conjugates to improve tumor specificity. These conjugates bind to tumor-specific antigens and may allow for precise delivery of either fluorescent markers or cytotoxic agents to glioma cells, enabling both visualization and targeted therapy. Recent preclinical studies have shown promise in targeting glioma-specific markers, suggesting future integration of these biologics with FGS platforms [137].

### 5.3. Current Clinical US Trials and Therapeutic Adjuncts

The FDA approval of 5-ALA led to many additional clinical trials for FGS, many of which are active and/or still recruiting (Table 2). The aim of these trials can be split into two categories: visualization-oriented trials and therapeutic-oriented trials. Several of these trials evaluate investigational uses of fluorophores in therapeutic settings, including early-phase studies of 5-ALA combined with sonodynamic therapy and novel agents such as Photobac and Pentalafen.

Visualization-oriented trials are seeking to expand indications for FGS. One clinical trial at the Icahn School of Medicine at Mount Sinai is exploring the use of 5-ALA in treating head and neck cancers [138]. At Dartmouth-Hitchcock Medical Center, there is a study investigating the use of 5-ALA in multiple primary and metastatic brain tumors including HGG, recurrent HGG, LGG, meningioma, and brain metastases [139]. There are also many clinical trials considering other fluorophores. A group at the University of Pennsylvania is testing the efficacy of SWIG in all patients with CNS tumors [140]. At Dartmouth-Hitchcock Medical Center, fluorescein is being evaluated in HGG and LGG [141]. Additionally, there are multiple visualization technologies that are currently being examined in clinical trials. A recently completed clinical trial at the University of Pittsburgh utilized a modified NICO^®^ Myriad handpiece (Stryker Corp., Kalamazoo, MI, USA) with a blue-light attachment for better visualization of fluorescent tumor tissue that is correlated with histopathology from tissue specimens harvested at the time of surgery. The NICO handpiece known as the SPECTRA was utilized in combination with the operative microscope or exoscope. The CONVIVO confocal endomicroscope is being studied in vivo for identifying HGG tumor tissue type using fluorescein [142]. Finally, bioptic loupes are being tested against optical microscopes for their ability to accurately identify fluorescing tumor tissue [143].

Artificial intelligence (AI) may soon augment fluorescence-guided surgery by automating intraoperative decision-making. Machine learning algorithms trained on intraoperative imaging and histopathologic datasets could assist in delineating tumor margins, identifying subtle fluorescence, or predicting tumor infiltration zones in real time. Early research supports the feasibility of integrating AI into fluorescence-based workflows [144].

## 6. Conclusions

FGS is a promising innovation within neurosurgical oncology for enhancing intraoperative visualization to improve the extent of resection and patient outcomes for glioma surgery. Adoption has increased dramatically in recent years and appears poised to continue increasing across centers doing brain tumor surgery. Ongoing trials hold promise for expanding indications for FGS as well as evaluating the possible therapeutic efficacy of fluorophores when used in combination with acoustic and light energy.

Despite growing clinical adoption in high-resource settings, global implementation of FGS remains limited by several barriers. These include the high cost of agents like 5-ALA, limited access to compatible imaging equipment in low- and middle-income countries (LMICs), and a lack of formalized training in fluorescence-based neurosurgical techniques. Overcoming these disparities will require coordinated global efforts, including the development of cost-effective portable imaging platforms and international training programs that leverage virtual surgical education and mentorship.

As research continues to expand and technology evolves, FGS is poised to remain a cornerstone in the neurosurgical management of gliomas. Its integration into standard practice reflects not only its clinical utility but also its potential to transform brain tumor surgery into a more precise, data-driven, and patient-tailored discipline. With continued innovation and investment, fluorescence-guided techniques hold the promise to elevate surgical outcomes for patients around the world.

## Figures and Tables

**Figure 1 cancers-17-01837-f001:**
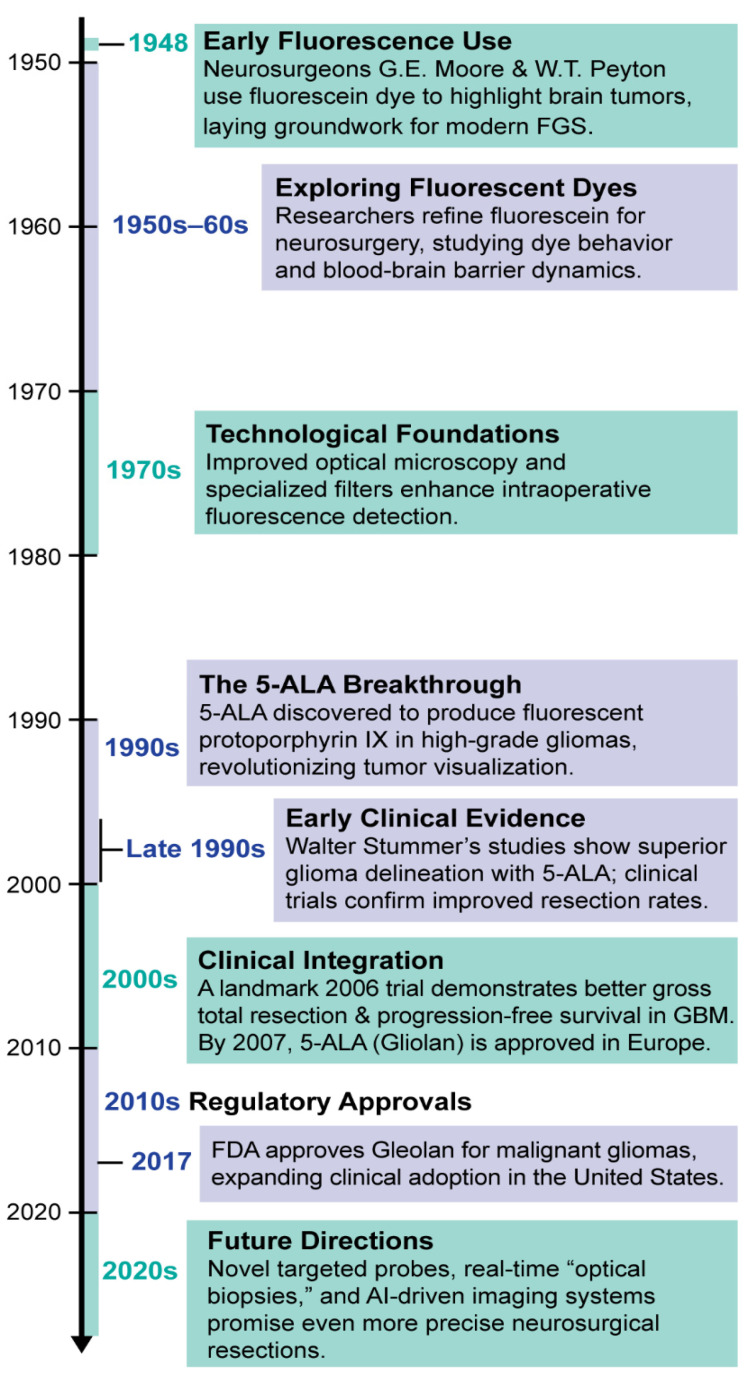
Key historical milestones in the development of fluorescence-guided surgery (FGS) in neurosurgery. The timeline highlights pivotal advancements, including the introduction of 5-ALA and fluorescein sodium, as well as major regulatory approvals and technological innovations that have shaped the clinical adoption of FGS techniques.

**Figure 2 cancers-17-01837-f002:**
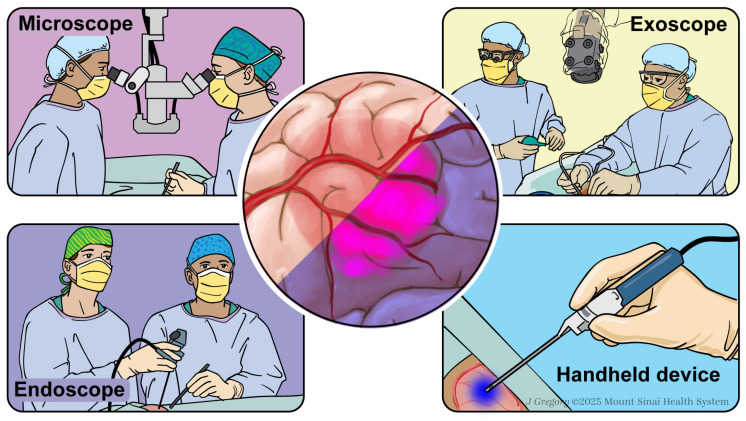
Contemporary applications of fluorescence-guided surgery (FGS) in neurosurgery. The illustration highlights the primary visualization platforms used intraoperatively, including surgical microscopes, exoscopes, endoscopes, and handheld devices. These modalities enable real-time identification of fluorescent tumor tissue, enhancing surgical precision and resection outcomes.

**Table 1 cancers-17-01837-t001:** Key Fluorophores Used in Glioma Surgery.

Fluorophore	Tumor Specificity	Depth of Penetration	GTR Rate	PFS (Months)	OS (Months)	Cost	Key Limitations
5-ALA	High	~1 to 2 mm	65–75%	6.8	15–20	$$$	Phototoxicity, limited depth
Fluorescein	Moderate	~1 mm	~60 to 70%	~9.2	~15	$	Non-specific, absorption interference
ICG (SWIG)	Low–Moderate	Deep (NIR, ~5 to 10 mm)	Not well defined	N/A	N/A	$$	Non-specific uptake, experimental

ICG: Indocyanine Green; SWIG: Second-Window Indocyanine Green; GTR: Gross Total Resection; PFS: Progression-Free Survival; OS: Overall Survival. Summary of key fluorophores used in glioma surgery, comparing regulatory status, optical characteristics, resection outcomes, and clinical limitations. Data reflect reported ranges from major trials and meta-analyses where available. The relative costliness of each compound is described as low ($), moderate ($$), or high ($$$).

**Table 2 cancers-17-01837-t002:** Ongoing and Recent Clinical Trials Involving Intraoperative Fluorescence Imaging and Therapy in Brain and Head/Neck Tumors.

Title	Clinical Trial Number	Sponsor	Indication	Fluorescence Type	Enrollment	Phase	Company
Second Window Indocyanine Green for All Nervous System Tumors	NCT05746104	Abramson Cancer Center at Penn Medicine	All CNS Tumors	ICG	105	1	TumorGlow (Pennsylvania, USA)
Study to Evaluate 5-ALA Combined With CV01 Delivery of Ultrasound in Recurrent High-Grade Glioma	NCT05362409	Alpheus Medical, Inc.	HGG	5-ALA	48	1	Alpheus Medical, Inc. (Minnesota, USA)
Diagnostic Performance of Fluorescein as an Intraoperative Brain Tumor Biomarker	NCT02691923	David W. Roberts, Dartmouth-Hitchcock Medical Center	HGG & LGG	Fluorescein	30	2	-
Sonodynamic Therapy in Patients With Recurrent GBM	NCT06039709	Shayan Moosa, MD, University of Virginia	rGBM	5-ALA	11	1	-
A Study of Sonodynamic Therapy Using SONALA-001 and Exablate 4000 Type 2.0 in Subjects With Recurrent GBM	NCT05370508	SonALAsense, Inc.	rGBM	5-ALA	44	1 and 2	SonALAsense, Inc. (California, USA)
A Phase 2 Study of Sonodynamic Therapy Using SONALA-001 and Exablate 4000 Type 2.0 in Patients With DIPG	NCT05123534	SonALAsense, Inc.	DIPG	5-ALA	27	1 and 2	SonALAsense, Inc.
ALA-Induced PpIX Fluorescence During Brain Tumor Resection	NCT02191488	David W. Roberts, Dartmouth-Hitchcock Medical Center	HGG, LGG, rHGG, Mets, Meningioma	5-ALA	540	1	-
Study of Sonodynamic Therapy in Participants With Recurrent High-Grade Glioma	NCT04559685	Nader Sanai, St. Joseph’s Hospital and Medical Center, Phoenix	rHGG	5-ALA	30	1	-
The Role of 5-Aminolevulinic Acid Fluorescence-Guided Surgery in Head and Neck Cancers: a Pilot Trial	NCT05101798	Alfred-Marc Iloreta, Icahn School of Medicine at Mount Sinai	Recurrent Head, Neck, or Skull Base	5-ALA	26	2	-
Loupe-Based Intraoperative Fluorescence Imaging	NCT04780009	Guoqiang Yu, University of Kentucky	GBM and AA	Fluorescein and 5-ALA	30	Observational	-
Evaluation of the CONVIVO System	NCT05139277	Linton T. Evans, Dartmouth-Hitchcock Medical Center	HGG, GBM, Mets, Meningioma, Acoustic Neuroma, Pituitary Adenoma	Fluorescein	30	Pre-Clinical	Zeiss (Oberkochen, Germany)
Intracavitary Photodynamic Therapy as an Adjuvant to Resection of Glioblastoma or Gliosarcoma Using IV Photobac	NCT05363826	Photolitec LLC	GBM Gliosarcoma	Photobac	30	1	Photolitec LLC (New York, USA)
A Dose-escalation Clinical Study of Intraoperative Photodynamic Therapy of Glioblastoma	NCT05736406	Hemerion Therapeutics	Newly diagnosed GBM	Pentalafen	12	1	Hemerion Therapeutics (Villeneuve-d’Ascq, France)

A summary of the key clinical trials evaluating various intraoperative fluorescence agents and technologies that include ICG, 5-ALA, and fluorescein in the surgical management of nervous system tumors. Trials are listed by indication, fluorescence type, sponsor, enrollment size, and phase.

## Data Availability

No new data were created.

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
