# Peer review of "Fluorescence-Guided Surgery for Gliomas: Past, Present, and Future"

_cancers, 2025, doi:10.3390/cancers17111837_

Round 1
Reviewer 1 Report
Comments and Suggestions for Authors
This is an interesting and well written review. A few specific comments are listed below.
89 Can the authors confirm the regulatory status if the other agents discussed in this review in the US and elsewhere. (e.g. “the other agents have received FDA clearance but not specifically for neurosurgical resections, so their use here remains off-label in the US. In Europe….)
178 word “orthopaedics” is repeated twice
366 gadolinium is mentioned here but there is no context. Is it used as a fluorophore or as a MR contrast agent? I suspect that the authors are referring to a postoperative scan but it is unclear since SWIG is an intraoperative modality and intraoperative MR is relatively rare.
403-408 Avoid using abbreviations and specify locations for the various sites Mount Sinai, Dartmouth, Penn.
411 – Use standard notation when referring to specific products. e.g. NICO (Stryker Corp., Kalamazoo MI)
Any adverse effects of the use of the various fluorophores should be briefly discussed.
The authors may wish to speculate on the possible use of antibody-drug conjugates in conjunction with some of the fluorophores (e.g. 10.3389/fonc.2021.718590).
The paper could benefit significantly if it included some speculation on the possible use of AI in conjunction with FGS (e.g. doi.org/10.1038/s41698-024-00699-3).
Reviewer 2 Report
Comments and Suggestions for Authors
This review article provides a comprehensive examination of FGS in the management of gliomas, with a primary focus on glioblastoma. The authors chronicle the historical development of FGS, highlight the clinical use of key fluorophores—particularly 5-ALA, ICG, and fluorescein—and describe advances in visualization technologies and therapeutic applications such as photodynamic and sonodynamic therapies. The manuscript also discusses the widespread adoption of 5-ALA FGS in neurosurgical practice, emerging dual-dye approaches, and ongoing clinical trials aimed at optimizing tumor visualization and treatment efficacy. By synthesizing current evidence and technological progress, the review outlines both the clinical impact and future potential of FGS in improving resection extent and patient outcomes.
Overall, the manuscript would benefit from greater critical analysis of the limitations associated with FGS, such as false positives, limited tissue penetration, and the specificity of fluorophores in heterogeneous tumor margins. A more balanced discussion of the comparative performance and cost-effectiveness of 5-ALA versus other agents (e.g., fluorescein, ICG) would strengthen the clinical relevance. Additionally, integrating more quantitative data on clinical outcomes (e.g., progression-free or overall survival across studies) and clearly distinguishing between established clinical evidence and experimental/preclinical findings would enhance clarity. Future directions could more explicitly address barriers to global adoption and implementation, particularly in low-resource settings. Structurally, a consolidated summary table of key fluorophores, visualization methods, and clinical trial outcomes would improve readability.
Round 2
Reviewer 2 Report
Comments and Suggestions for Authors
thanks for having addressed my comments